# A Differentiable Recipe for Learning
# Visual Non-Prehensile Planar Manipulation

**Bernardo Aceituno[1], Alberto Rodriguez[1], Shubham Tulsiani[2], Abhinav Gupta[2], Mustafa Mukadam[2]**

[1]Massachusetts Institute of Technology, [2]Facebook AI Research

**Abstract:** Specifying tasks with videos is a powerful technique towards acquiring novel and general robot skills. However, reasoning over mechanics and dexterous interactions can make it challenging to scale learning contact-rich manipulation. In this work, we focus on the problem of visual non-prehensile planar manipulation: given a video of an object in planar motion, find contact-aware robot actions that reproduce the same object motion. We propose a novel architecture, Differentiable Learning for Manipulation (DLM), that combines video decoding neural models with priors from contact mechanics by leveraging differentiable optimization and finite difference based simulation. Through extensive simulated experiments, we investigate the interplay between traditional model-based techniques and modern deep learning approaches. We find that our modular and fully differentiable architecture performs better than learning-only methods on unseen objects and motions. https://github.com/baceituno/dlm.

**Keywords:** Manipulation, Visual learning, Differentiable optimization

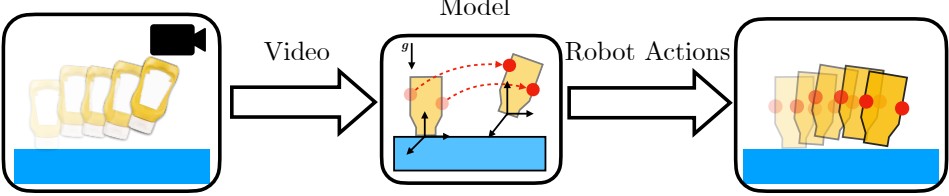

Figure 1: We tackle the problem of visual non-prehensile planar manipulation where given a pre-segmented video of an object in planar motion (left) the goal is to find the robot actions (middle) to reproduce the same object motion (right). In learning for manipulation, we study the role of structure, intermediate representations, and end-to-end differentiation.

## 1 Introduction

Dexterous manipulation tasks can be easily described through videos and have the potential to scale robot learning, but decoding relevant information from them remains a challenging problem. While recent approaches employing deep learning from images and videos have seen remarkable success in various prehensile robotic tasks [1, 2, 3, 4], there has been limited progress in the direction of non-prehensile manipulation. An underlying struggle lies in building representations that can reason over object geometry, rigid-body mechanics, and the combinatorial decisions of choosing contact-modes. Mechanics representations and priors are ubiquitous in model-based techniques and thus have been able to demonstrate working solutions [5, 6, 7, 8]. However, these methods can be computationally expensive, are unable to exploit past experience, and require hand engineering, as they assume access to known object shape, pose, and desired trajectory.

Can bringing mechanics representations and priors into deep learning models aid in decoding videos and learning dexterous manipulation? We study this question on the problem of video-based non-prehensile planar manipulation. Here, given a video of an object in planar motion in some environment (i.e. input video), the goal is to find contact and force trajectories for the robot fingers (i.e. output robot actions) that when executed will reproduce the same object motion in the same environment. This problem is illustrated in Fig. 1. Our approach is to build intermediate representations based on mechanics that serve as a glue in combining upstream neural models that decode the video with downstream differentiable priors that solve for robot actions. We limit the problem scope to planar settings, with the input video being pre-segmented,

5th Conference on Robot Learning (CoRL 2021), London, UK.

and the robot having point-fingers (standard in robot manipulation literature [9, 10, 11, 6, 8]). These simplifications allow us to assess the impact of structured learning without the confounding effects of sensor or actuator noise, and enable us to make a reasonable first step towards general approaches that would be able to learn dexterous manipulation by watching large scale human videos [12].

We present an architecture, Differentiable Learning for manipulation (DLM) (illustrated in Fig. 2), that works in three stages and can be progressively trained on each stage: (1) a neural model that derenders the input video to mechanical parameters i.e. object shape, object trajectory, discrete contact-mode decisions, and allowable forces between robot fingers and object, (2) a differentiable convex optimization module based on cvxpylayers [13, 14] that solves inverse dynamics from the mechanical parameters to find contact and force trajectories for the robot fingers, and finally (3) a differentiable contact-mechanics simulation module based on lcp-physics [15] and finite-differencing that executes the robot actions to simulate the object motion. We train the first two stages with supervision from an expert contact trajectory optimizer [6] while the third stage is trained end-to-end over the desired object trajectory.

To investigate the role structure plays in learning, we compare our approach with various architectures including neural models without any mechanics priors on visual non-prehensile planar manipulation tasks. In our experiments, we train the third stage of DLM on a dataset that is smaller relative to the training dataset used for the first two stages to resemble a more realistic setting where pre-training is done on a large simulation dataset with easy to obtain supervision and fine-tuning is done on a smaller real-world dataset that is expensive to collect and harder to label. We find that our structured model is able to outperform other learning-based architectures on unseen objects and motions, while being more computationally efficient compared to the non-learning expert.

## 2 Related Work

Prior work in visual manipulation has involved learning predictive models from videos of a robot interacting with its environment. These models are used to find actions with simulation roll-outs, in a model-predictive fashion [16, 17, 1, 18, 19, 20, 21]. These approaches have been successful at solving prehensile manipulation problems where the dynamics are hard to model. When the dynamics have a known model, while these approaches remove the need to analyze visual data, mechanics or geometry, they struggle in generalizing to different tasks and geometries and can be data inefficient. Our work aims to address this limitation, under the planar non-prehensile manipulation task, by learning mechanical parameters from video and optimizing robot finger actions with a known rigid-body mechanics model, instead of learning a fully predictive model.

On the other side of the spectrum, model-based techniques leverage the known contact-mechanics that govern the manipulation process for planning or control. The scope of these techniques is often focused in: (i) the planning/control of primitives [22, 23, 9], which can be combined sequentially to complete a task [7, 24], or (ii) in the optimization of general skills [25, 5, 6], which solve inverse rigid-body dynamics to solve a task. These approaches have the potential to provide a certification of robustness [10, 26] and versatility to operate with different robot kinematics and task specifications. However, they also depend on accurate state-estimation and control to be implemented. Moreover, solving contact mechanics involves non-smooth optimization or large combinatorial search. This prevents the scalability of these methods to complex geometries and large time horizons. The main focus of the state-of-the-art is in planar manipulation tasks over sagittal settings, where friction and contact forces are the primary source of challenges [5, 27, 6, 28].

Recent research has tried to bridge the gap between deep learning and model-based techniques by introducing differentiable computational techniques as part of the learning pipeline. The first relevant line of work involves the introduction of differentiable optimization solvers [29, 30, 13], which introduce a parametric convex optimization program as a differentiable layer in a neural network. We leverage one of these solvers [13] to solve a quadratic program that takes as input mechanical parameters from a task (derendered from video) and outputs robot finger trajectories within our network. The second relevant line of work comes from the implementation of differentiable contact-mechanics simulators [15, 31], which execute a parametric physical simulation and allow to back-propagate from a measured output. We modify an existing differentiable simulator [15], using finite-differences, to handle intermittent contact dynamics and evaluate the error when executing the learned robot finger trajectories as part of a loss function.

## 3 Visual Non-Prehensile Manipulation Inference

In this section, we define our problem of inferring manipulation actions from a video which serves as a task specification. We limit our focus to the context of 2D non-prehensile manipulation given 2D

pre-segmented videos of desired object motion. In the discussion section, we elaborate on possible avenues to extending our formulation and approach to 3D, real videos, and real robots. We start by defining the main components of our problem, its assumptions, and our notation:

1. **Task:** a video $\mathcal{V}$ with $T$ frames showing a sequence of object poses. The tasks involves a set of object poses $\mathbf{r}(t) \in SE(2)$, sampled in $T$ time-steps $t$.

2. **Object:** a polygonal rigid-body $\mathcal{O}$ with mass matrix $\mathbf{M}$ and $N_F$ facets. Each facet $\mathbb{F}_f$ has a corresponding friction cone $\mathcal{FC}_f$.

3. **Action Space:** a set of $N$ point-fingers[1] moving freely around space. We describe the position of finger $c$, at time-step $t$, as $\mathbf{p}_c(t)$. The contact interaction between the object and the finger at $\mathbf{p}_c(t)$ applies a force $\lambda_c(t)$.

4. **Environment:** a polygonal environment described as planes with friction cones $\mathcal{FC}_e$. The contact interaction between the object and the environment occurs at the contact point $\mathbf{p}_e(t)$ and results in a reaction force $\lambda_e(t)$.

We show a high-level representation of our problem in Fig. 1. One of the challenges in solving this problem come from interpreting video data, in order to extract an implicit representation of the object shape, the object motion, and finding the appropriate contact interactions between the robot fingers and the object. Specifically, finding a contact interaction is divided in two steps: (i) select a sequence of *contact modes*, indicating where each contact is applied at each time-step, and (ii) *invert the dynamics* to resolve the exact contact locations and forces to apply. In this context, inverse dynamics (ID) is an optimization problem that receives an object motion and outputs forces and locations, formulated in the form:

$$\textbf{ID:} \qquad \min_{\mathbf{p}, \boldsymbol{\Lambda}} \quad \sum_{t=0}^{T} J_{p,\lambda}^{ID}(t) \tag{1}$$

$$\text{subject to:} \quad \mathbf{M}\ddot{\mathbf{r}}(t) + \mathbf{G}(\mathbf{r}) = J(\mathbf{r})^T \Lambda(t), \quad \mathbf{p}_c(t) \in \mathbb{F}_c(t), \quad \lambda_c(t) \in \mathcal{FC}_c(t), \quad \lambda_e(t) \in \mathcal{FC}_e(t) \tag{2}$$

where $J_{p,\lambda}^{ID}(t)$ is a cost function that typically penalizes actuation efforts, $\mathbf{G}(r)$ represents gravity and inertial effects, and $J(r)$ is a Jacobian mapping contact forces $\Lambda = [\lambda_1, ..., \lambda_N, \lambda_e]$ into wrenches. Solving **ID**, however, assumes knowledge of the exact shape of the object and the contact modes to be applied [32, 27, 7], encoded in the facets $\mathbb{F}_c(t)$ and friction cones $\mathcal{FC}_c(t)$. Solving this problem without assuming known contact modes is often intractable and a significant body of research has focused on studying it [25, 33, 6]. Our approach addresses this by extracting object shape and contact modes (as facets and friction cones) as parameters through a deep neural model, which can be trained with ground-truth labeled data and augmented with self-supervision through differentiable optimization and differentiable simulation.

## 4 Differentiable Visual Non-Prehensile Manipulation

In this section, we present our approach Differentiable Learning for Manipulation (DLM). Each subsection describes a stage of the model and provide technical and implementation details. Our approach leverages deep neural networks, differentiable optimization, and simulation, to construct the full pipeline, illustrated in Fig. 2.

### 4.1 Mechanical Derendering

One challenge of visual manipulation is to decode geometry and motion from video. While pre-segmented videos can be obtained from realistic videos with state-of-the-art computer vision tools [34], it is challenging to translate explicit task information into an implicit latent space. This is due to discrete variables such as facets, vertices, and intersections. Hence, the first stage of our model encodes the video frames $\mathcal{V}$ into an implicit latent space $\mathcal{L}^\mathcal{V}$. To achieve this, we first encode each frame $I_k$ through a set of convolution layers $CNN(\cdot)$ and combine these embeddings with a Long Short-Term Memory (LSTM) layer, to retain their temporal relation, as:

$$L^\mathcal{V} = LSTM(CNN(I_1), CNN(I_2), ..., CNN(I_T)) \tag{3}$$

---

[1]The use of point-fingers does not account for robot kinematics. While this is a limitation, it is a standard approach to evaluate planning and learning pipelines for manipulation problems (see [9, 28, 10, 26, 6, 8]).

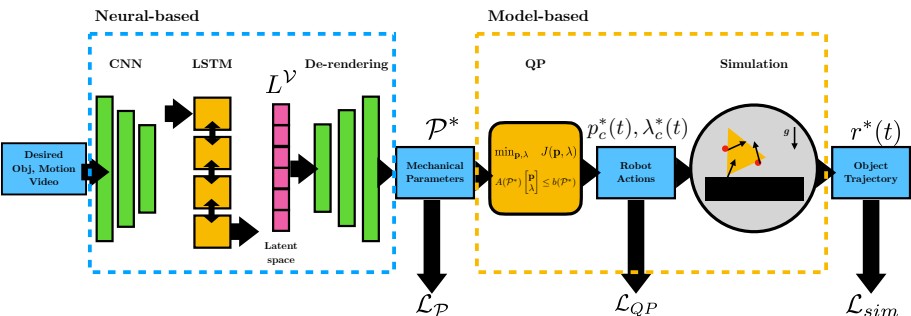

Figure 2: Our proposed fully differentiable pipeline, DLM (Differentiable Learning for Manipulation). In contrast to approaches that represent policies as feed-forward neural networks, our approach leverages model-based priors from mechanics to process mechanical parameters via a differentiable quadratic program and evaluate the resulting policy through a differentiable simulator.

Given this encoding, we find all the parameters required for **ID** with differentiable optimization, in similar fashion to [35]. This set of mechanical parameters [6] includes: (i) the Jacobian matrix $J(\mathbf{r})(t)$, (ii) the location of external contacts $\mathbf{p}_e(t)$, and (iii) parameters that implicitly encode contact modes $\mathbb{F}_c$, $\mathcal{FC}_c$, and $\mathcal{FC}_e$. We derender this set of mechanical parameters $\mathcal{P} = \{\mathbf{r}(t), J(\mathbf{r})(t), \mathbb{F}_c(t), \mathcal{FC}_c(t), \mathbf{p}_e(t), \mathcal{FC}_e(t)\}$ through an MLP[2] and the loss function:

$$\mathcal{P}^*(t) = MLP^{\mathbf{P}}(L^{\mathcal{V}})(t), \forall t, c, \quad (4)$$

$$\mathcal{L}_{\mathcal{P}} = ||\mathcal{P}^*(t) - \mathcal{P}(t)||_2^2 \quad (5)$$

As illustrated in Fig. 3 these parameters are an implicit representation of the finger contact modes and the local object geometry for a given task, since the contact mode only encodes which facet is in contact and what is the friction cone.

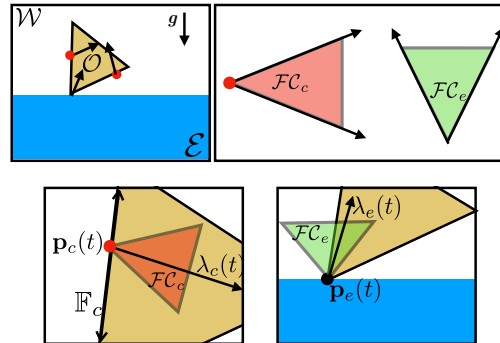

Figure 3: Interactions in our setup (top-left) and the mechanical parameters from the scene (bottom). Different contact modes are encoded implicitly via the friction cones and the object facet at each contact (top-right). Note that the input video only demonstrates the object moving without any robot actions.

### 4.2 Differentiable Inverse Dynamics

Given the derendered parameters, we recast **ID** as the following quadratic program which solves for robot actions $p_c(t), \lambda_c^*(t)$:

$$\textbf{QP:} \quad \min_{\mathbf{p}_c, \lambda, \epsilon} \quad J_{QP} = \sum_{t=0}^{T} ||\Lambda_c(t)||_2^{Q_\lambda} + ||\ddot{\mathbf{p}}_c(t)||_2^{Q_p} + q|\epsilon(t)|$$

subject to: $\mathbf{M}\ddot{\mathbf{r}}^*(t) = J(\mathbf{p}, \mathbf{r})^{*T}\Lambda(t) - \hat{\mathbf{G}}(\mathbf{r}^*) + \epsilon(t), \ \lambda_c(t) \in \mathcal{FC}_c^*(t), \ \lambda_e(t) \in \mathcal{FC}_e^*(t), \ \mathbf{p}_c(t) \in \mathbb{F}_c^*(t)$ (6)

Under this linearization, **QP** has the property of being a convex optimization problem [36]. This has a few benefits: (i) its solution is always the global optima, (ii) if $\mathcal{P}^*(t) = \mathcal{P}(t)$ then its solution is guaranteed to match the ground-truth optima, and (iii) **QP** can be added as a differentiable layer of our model [29, 13], which inputs $\mathcal{P}^*$ and outputs the optimal $\mathbf{p}_c^*, \lambda_c^*, \epsilon$ for such parameters. There are a few considerations to make **QP** a differentiable layer. In particular, **QP** must have a solution–or be feasible– for any choice of parameters $\mathcal{P}^*(t)$ [13]. To make the problem always feasible, we relax the dynamics constraints by adding a slackness term $\epsilon(t)$ and penalize it in the loss function $J_{QP}$, with weight $q$. After solving **QP**, we supervise its solution with ground-truth labels, under known parameters, with the loss

$$\mathcal{L}_{QP} = q|\epsilon| + ||\mathbf{p}_c^* - \mathbf{p}_c||_2^2 + ||\lambda_c^* - \lambda_c||_2^2, \quad (7)$$

which drives the model towards a set of actions that are both consistent with the physical structure, such that $\epsilon(t) \to 0$, the ground-truth physical parameters $\mathbf{p}_c^*, \lambda_c^* \to \mathbf{p}_c, \lambda_c$. After training, we set $\epsilon = 0$ to ensure consistency.

---

[2]Facets $\mathbb{F}$ are represented by two vertices in the world frame, while friction cones $\mathcal{FC}$ are represented with two rays originating at the contact point.

## 4.3 Evaluation

A drawback of learning directly from the action space is the dependence on *ground-truth* finger trajectories to assess the generality of the learned model. The existence of multiple solutions, e.g. as depicted in Fig. 4, makes it ambiguous to assess a model with unseen labeled data. To resolve this ambiguity in the evaluation, we simulate the task using inferred actions $p_c^*(t), \lambda_c^*(t)$. We then compare the simulated object trajectory with the desired object trajectory, and use this error as a metric [37]. Algebraically, a simulator is represented by

$$\mathbf{r}^*(t) = Sim(\mathbf{p}_c^*(t)) \begin{cases} \mathbf{r}(t+1) = \mathbf{r}(t) + \Delta T \dot{\mathbf{r}}(t), \\ \dot{\mathbf{r}}(t+1) = \dot{\mathbf{r}}(t) + \Delta T \ddot{\mathbf{r}}(t), \\ \mathbf{M}\ddot{\mathbf{r}}(t) + \mathbf{G}(\mathbf{r}) = J(\mathbf{r})^T \mathbf{\Lambda}^f(t) \end{cases} \tag{8}$$

and:

$$\mathbf{\Lambda}^f = [\lambda_1^f(\mathbf{p}_1^*(t)), ..., \lambda_N^f(\mathbf{p}_N^*(t)), \lambda_e^f(\mathbf{p}_e^*(t))]$$

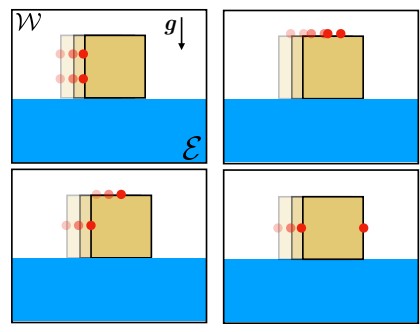

Figure 4: Example of four different valid solutions to a simple manipulation task of pushing a block from left to the right. Since the video does not show what actions to follow, our model may output any of these actions depending on its supervision.

is a functional representation of the contact forces. These forces need to be determined based on the inertial properties of the system. To resolve these forces we solve a Linear Complementary Problem (LCP), based on [15], which outputs the corresponding contact forces between the object, fingers, and the environment. These contact forces are activated discretely as:

$$\lambda_k^f((\mathbf{p}_k^*(t))) = \delta(D(\mathbf{p}_k^*(t), \mathbf{r}(t))), \quad D(\mathbf{p}_k^*(t), \mathbf{r}(t)) \geq 0, \delta(x) = \begin{cases} 1, x=0 \\ 0, x>0 \end{cases} \tag{9}$$

where $D(\mathbf{p}_k^*(t), \mathbf{r}(t))$ is the distance between point $\mathbf{p}_k^*(t)$ and the object at pose $\mathbf{r}(t)$. Similar relations are used to represent the friction cones and contact modes, which we omit for simplicity. Then, we use the simulation for the evaluation metric using the loss function

$$\mathcal{L}_{sim} = ||Sim(\mathbf{p}_c^*(t)) - \mathbf{r}(t)||_2 \tag{10}$$

where $\mathbf{p}_c^*(t)$ are finger trajectories obtained by solving **QP**. It is important to note that $\mathcal{L}_{sim}$ is not an injective mapping from the finger trajectory error $||\mathbf{p}_c^*(t) - \mathbf{p}_c(t)||_2$. Therefore, having actions near the ground-truth might not necessarily lead to a lower $\mathcal{L}_{sim}$.

## 4.4 Supervision via Simulation

Having access to the loss function $\mathcal{L}_{sim}$ can also tune our model subject to ground-truth physics. This reduces (or potentially removes) the burden of having labeled data over the parameters $\mathcal{P}(t)$ and the finger-trajectories $\mathbf{p}_c(t)$, leading to a fully self-supervised approach. Training our model with $\mathcal{L}_{sim}$ requires the function $Sim(\cdot)$ (i.e. the simulator) to be differentiable in all its domain. However, the function $\delta(\cdot)$ only returns gradients at the origin. While this is a pervasive problem, many researchers have successfully included contact mechanics as differentiable functions by approximating $\delta(\cdot)$ with a smooth function [25]. In our case, we approximate $\delta(\cdot)$ as the smooth function:

$$\delta(D(\mathbf{p}_k(t)) \approx \sigma(-D(\mathbf{p}_k(t), \mathbf{r}(t)), \kappa)$$

where $\sigma(\cdot, \kappa)$ is a sigmoidal function with factor $\kappa$. This approximation can be made arbitrarily tight as $\kappa \to \infty$. With this smoothing, the simulator gradients are computed via finite differences as:

$$\nabla \mathbf{r}^*(t) \approx (Sim(\mathbf{p}_k^*(t) + h_p) - Sim(\mathbf{p}_k^*(t) - h_p))/2h$$

where $h_p = h\mathbf{I}_{3\times3}$. While automatic differentiation is a more efficient approach to obtain gradients, we found it incompatible with the smoothed contact model of our current implementation. We leave this enhancement for future work. Minimizing $\mathcal{L}_{sim}$ drives the learned finger trajectories to push the object through the desired object trajectory. A consequence of this approximation is that $\delta(\cdot) \approx \sigma(\cdot)$ leads to effects such as contact at distance and small penetration, although these are attenuated for a large $\kappa \gg 1$ in practice.

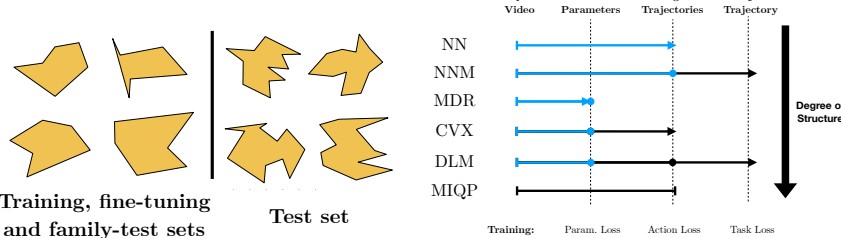

Figure 5: **Left:** Examples shapes of objects in our different datasets. **Right:** Architectures used to evaluate our framework. Blue segments are stages in the architecture with a neural model, black segments are differentiable model-based stages. Dots represent points with labeled data used for pre-training and arrow ends are the final loss function for the full architecture. The MIQP approach has privileged access to all the parameters of the problem.

## 5 Experiments

We implement all models in PyTorch [38] with Python 3. To generate our datasets, we use MATLAB R2020b and solve optimization problems with the Gurobi [39] solver. For all experiments, we use Adam as our network optimizer and CVXPY layers [14, 40] as differentiable solver for **QP**. In all experiments we train over each loss function for 100 iterations with a learning rate of $10^{-4}$. We implement our simulator on top of the LCP-based scheme in [15] by modifying it to accommodate our problem setting. We run all training, testing, and timing experiments on a Macbook Pro with 2.9 GHz 6-Core Intel CPU.

**Datasets:** We built four datasets of non-prehensile manipulation tasks in the *sagittal*[3] plane with randomized trajectories in $SE(2)$, all ground truth robot actions are found by solving Contact-Trajectory Optimization (CTO) [6]:

- *Training set* of 20 tasks with randomly generated polygonal objects of 4 to 6 facets, including all mechanical parameters and robot actions for each task.
- *Fine-Tuning set* of 20 tasks with new random objects of 4 to 6 facets, with robot actions for each task.
- *Test set* of 40 tasks with new random objects of 12 facets, with robot actions for each task.
- *Family Test set* of 40 tasks with new random objects of 4-6 facets (same shape and task distribution as training and fine-tuning sets), with robot actions for each task.

Each task has a video of $T = 5$ RGB frames of size $50 \times 50$. We find optimal finger trajectories for each task using a Mixed-Integer Quadratic Program (MIQP) [6], for $N = 2$ point fingers. We set $\kappa = 0.5$, $q_\epsilon = 0.1$ and $dim(\mathcal{L}^{\mathcal{V}}) = 75$. We illustrate example objects we generate for each dataset in Fig. 5 (left). Also see Appendix for more details on data generation.

**Architectures:** We asses the performance of our approach, Differentiable Learning for Manipulation (DLM) against four different architectures and a model-based oracle (see Fig. 5).

- NN: The Neural Network (NN) architecture replaces **QP** in DLM, with a 3-layer MLP. This is akin to a pixel-to-actions style policy network [41, 3] and is trained to generate MIQP solutions with a loss function $\mathcal{L}_{NN} = ||\mathbf{p}_c^*(t) - \mathbf{p}_c(t)||_2^2$.
- NNM: The Neural Network Manipulation (NNM) architecture extends NN by also fine-tuning by minimizing $\mathcal{L}_{sim}$, akin to the examples used in [15]. This network is pre-trained with 100 iterations of minimizing $\mathcal{L}_{\mathcal{NN}}$.
- MDR: In the Mechanical DeRendering (MDR) architecture, DLM is cut off when it is trained to extract parameters from video by minimizing the loss function $\mathcal{L}_{\mathcal{P}}$. At test time robot actions are obtained by solving **QP**. This is akin to the approach used to learn integer solutions to mixed-integer programs [42, 43].
- CVX: In the ConVeX optimization (CVX) architecture, DLM is cut off after being trained by minimizing $\mathcal{L}_{QP}$. This network is pre-trained with 100 iterations of minimizing $\mathcal{L}_{\mathcal{P}}$.
- DLM: Our proposed approach is trained by minimizing $\mathcal{L}_{Sim}$ after it is pre-trained with 100 iterations of minimizing $\mathcal{L}_{\mathcal{QP}}$ which in turn is pre-trained with 100 iterations of minimizing $\mathcal{L}_{\mathcal{P}}$.
- MIQP (Oracle): A model-based solution to each manipulation task, solved via CTO [6] and provided with all ground-truth parameters (such as object shape and desired object trajectory in $SE(2)$).

---

[3] side-view of the scene with gravity pointing down in the image plane.

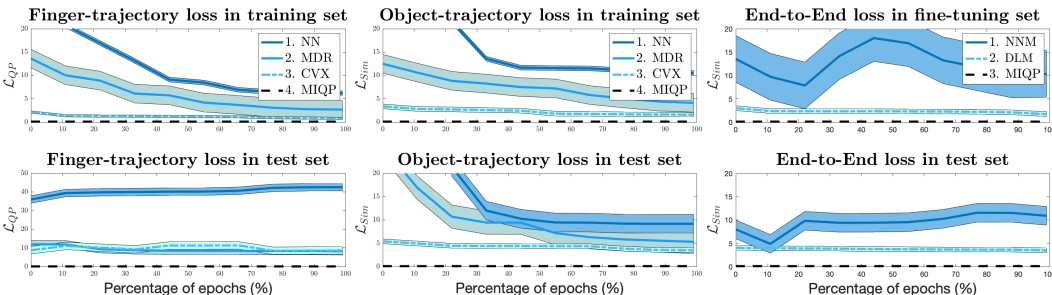

Figure 6: **Left:** In contrast to NN, MDR and CVX closely match the finger-trajectory solution from MIQP (Oracle). **Center:** While each network is trained to match finger trajectories from MIQP, the more structured networks (MDR and CVX) generalize better to unseen object trajectories and shapes. **Right:** Each network is trained by propagating gradients through the simulator. The DLM architecture performs better than the unstructured NNM without over-fitting.

Table 1: Object-Trajectory Loss and computation times (in seconds) for the different architectures on each dataset.

| Model | Training | Fine-Tuning | Test | Family test | Backward Pass (s) | Forward Pass (s) |
|---|---|---|---|---|---|---|
| NN | 8.5±0.5 | N/A | 9.5±2 | 10.0±1.5 | 0.05±0.01 | 0.06±0.02 |
| NNM | N/A | 15.0±5.00 | 11.0±2.00 | 15.0±5.0 | 40.00±5.00 | 0.06±0.02 |
| MDR | 4.5±2.0 | N/A | 7.5±2.5 | 4.5±2.0 | 0.10±0.01 | 0.11±0.02 |
| CVX | 2.0±0.5 | N/A | 4.0±0.5 | 3.5±0.5 | 0.23±0.01 | 0.11±0.02 |
| DLM | - | 1.5±0.5 | 3.5±0.5 | 3.5±1.0 | 40.00±5.00 | 0.11±0.02 |
| MIQP | - | - | - | - | - | 0.57±0.25 |

**Network details:** The $CNN(\cdot)$ and $DeConv(\cdot)$ operators represent 3 (de)convolution operations with 6, 12, and 24 channels. The $LSTM(\cdot)$ function is a 5-layer bi-directional recurrent LSTM. Each $MLP(\cdot)$ has 3 fully-connected hidden layers of equal size to their input. All the layers use ReLU activation functions. We add a 0.2 dropout and batch-normalization to each fully-connected layer of the network.

## 5.1 Learning Policies from Data

We first analyze the ability to infer finger trajectories from the ground truth data obtained via CTO.

**Finger trajectory loss:** We measure the ability of NN, MDR, and CVX methods to learn specific finger actions from the training set and evaluate their generalization to unseen data from the test set. We present the results of running each network for 100 epochs in Fig. 6 (left). The NN network consistently falls in a local minima. In contrast, the MDR and CVX result in lower training loss closer to the MIQP (Oracle) reference. CVX further refines the learned weights for more accurate predictions, since it is pre-trained with learned weights from MDR.

**Simulated object trajectory loss:** As we mention in Section IV.D, when we presented new data, the networks struggle to generalize finger trajectories to new object shapes. This is a consequence of each problem having multiple solutions. To resolve this ambiguity, we measure the *object-trajectory* loss by simulating the learned actions at each learning iteration. We show the object trajectory loss curves for each of our datasets in Fig. 6 (center). Fig. 6 (center) demonstrates how the inferred actions lead to better execution on unseen objects, despite little improvement on the finger-trajectory loss. We note how training with CVX not only refines the loss of MDR, but also prevents over-fitting. Examples of the qualitative performance of each model are shown in Fig. 7.

## 5.2 Learning Policies from Simulation

Next we study the impact of learning actions in a self-supervised fashion. We train DLM and NNM networks on the Fine-tuning set, leveraging the differentiability of our simulator. Both networks are pre-trained with the training set. We plot the resulting object trajectory loss in Fig. 6 (right). The DLM architecture decreases the test loss without overfitting on the Fine-tuning set, while the NNM network oscillates around a local minima. This test presents the most similarity to a real-world experiment where, without ground-truth labels on the parameters, the network learns to execute a video motion by iterating over experiments. Examples of qualitative performance of DLM model is shown in Fig. 7. We show an example of how this approach can be applied to real object, given a segmented video, in Fig. 1. Also see Appendix for results on shapes from the OmniPush dataset [44].

## 5.3 Comparative Analysis

Finally, we compare the final performance of the different networks in terms of object-trajectory loss and computation times (per data-point), shown in Table 1. Inference time in a neural model is smaller than

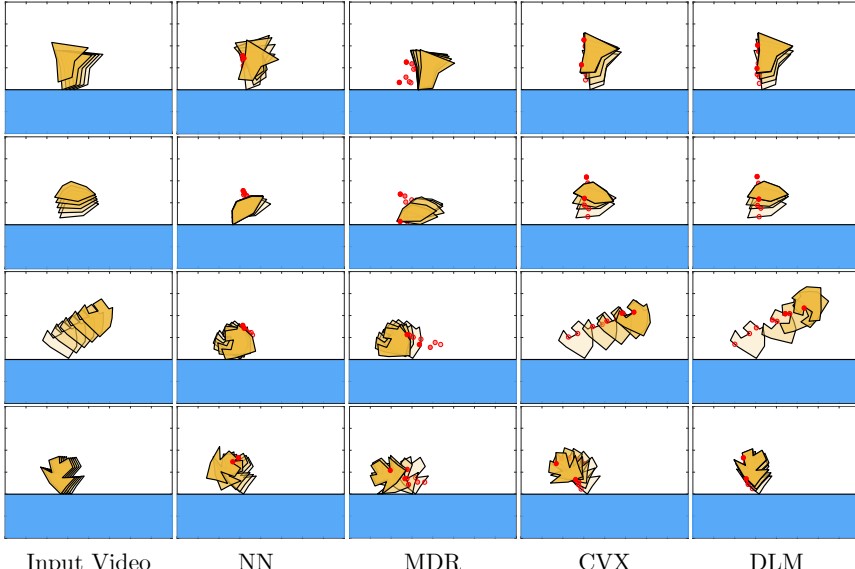

| Input Video | NN | MDR | CVX | DLM |

Figure 7: Qualitative examples of each network applied to four trajectories from the *family* (top two) and *test* (bottom two) sets. These shapes and motions are not seen during the training phase of each model. Networks with more embedded mechanical structure (MDR, CVX, DLM) tend to perform better than less structured ones (NN). The DLM pipeline provides a layer of self-supervision to the model which helps it to generalize better to unseen scenarios.

MIQP (Oracle) [6], as our model involves simple operations and solving **QP**. We note that training the DLM model is slow, due to the use of finite difference in the backward pass.

## 6 Discussion

This work explores the problem of visual non-prehensile planar manipulation, reconciling tools from model-based mechanics with deep learning. Our proposed Differentiable Learning for Manipulation (DLM) approach: (i) encodes the input video $\mathcal{V}$ in a latent vector $L^{\mathcal{V}}$, (ii) derenders mechanical parameters $\mathcal{P}^*(t)$ for the task, (iii) solves **QP** [29] to obtain robot finger actions $p_c^*(t), \lambda_c^*(t)$, and (iv) evaluates the performance of the model by simulating these actions. We train this model by minimizing $\mathcal{L}_{\mathcal{P}}$, in order to match the ground-truth parameters, and by minimizing $\mathcal{L}_{QP}$, in order to match the ground-truth actions. Moreover, we can self-supervise this approach with new data by back-propagating through the simulator [15] by minimizing $\mathcal{L}_{sim}$, without the need for ground-truth labels on parameters or actions. We assess this method by learning how to solve planar manipulation tasks given a pre-segmented video showing a desired object motion. Our experiments suggest that, when compared to fully neural architectures, our approach can generalize better to unseen tasks and shapes with the same amount of training data.

**Limitations:** The first limitation of this approach comes from the differentiable simulation scheme. Since our implementation uses finite differences, the slow backward pass hampers the scalability of our model. Similarly, the approximations required to make contact-mechanics differentiable also lead to undesired effects, such as rigid-body penetrations and contact at distance, which can compromise the quality of the results. A second limitation arises from our mechanical assumptions. Assuming that contacts occur at a set of points disregards tasks involving multiple or continuous contacts (e.g. pivoting against a wall). Moreover, the representation of the friction cones brings challenges for extending to 3D scenarios, as it cannot appropriately represent sliding contact. Finally, since our setup uses unconstrained point fingers, we are unable to represent more realistic robot kinematics.

**Future work:** The next step towards our vision is applying this framework in a real-world system, requiring the capability to handle real-world video data and robot kinematics. Our vision is to perform training on completely synthetic data generated with MIQP [6], fine-tuned for robustness with the simulator, and execute a task specified with a real-world video. Exploring faster simulation schemes will also be essential to scale this model to larger datasets. Techniques that explicitly resolve the contact-mechanics, through differentiable solvers, can alleviate this issue [15, 31]. Extending this framework to more complex manipulation tasks might require an implicit representation for multiple environmental contacts. Finally, moving to 3D tasks in $SE(3)$ is also possible by solving contact-trajectory optimization [6] in a 3D environment [6, 8] also leveraging recent advances in large-scale 3D differentiable simulation [31].

**Acknowledgments**

We thank Felipe de Avila Belbute-Pers for help with the LCP simulator and Hongkai Dai for helpful discussions. Part of this work was done while Bernardo Aceituno was an intern at Facebook AI Research. Bernardo Aceituno was supported by the Mathworks Engineering Fellowship.

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
