# OpenReview forum: "A Differentiable Recipe for Learning Visual Non-Prehensile Planar Manipulation"
_robot-learning.org/CoRL/2021/Conference — CoRL2021 Poster_

### Official Review · Reviewer_teuP · 2021-07-23

**Originality:** Good
**Technical Quality:** Good
**Clarity Of Presentation:** Good
**Impact:** 2

**Recommendation:**

Weak Reject: I recommend rejecting the paper, but will not argue for my recommendation if the majority of other reviewers have a different opinion.

**Summary:**

The authors suggested a new pipeline for learning 2D manipulation tasks. The paper is well written, however, the proposed method is limited to 2D cases. Even those 2D examples in the paper are not complicated. There is only one object and one blue floor. The only interaction between the object and the floor is friction and gravity. There is no comparison between the proposed method and other methods. The authors only compared different versions of their method.

**Issues:**

some issues:
1) Is the model (two parts) trained end-to-end or separately?

2) In the first row of Fig.7, two fingers are on one side of the object. The question is why the object can be raised.

The major problem of the paper is that the authors are working on simple toy examples. According to the authors themselves, the method cannot be easily generalized to 3D tasks. "Implementing this framework in a real-world system is paramount, requiring the capability to handle real-world video data and robot kinematics". But it is exactly what we should do.  In literature, people are using CNN+LSTM to handle real-world video data. As the authors mentioned, the mechanical parameters can be quite complicated for 3D tasks. It means that the generalization of the proposed method might be poor.

**Reviewer Expertise:**

Good: General knowledge of the area

**Strengths And Weaknesses:**

The paper is well written. The model seems to be promising.
However,

1) it is only evaluated with toy examples. I think that it is not easy to generalize the method for 3D or more complicated tasks.

2) The method also requires a differentiable simulator because the cost function is based on the simulator. It might work for these 2D examples. But it might not be easy to have a differentiable simulation for more complicated tasks.

3) There is no comparison with other methods.

**Summary Of Recommendation:**

Overall, the authors are solving some problems which might not be challenging enough as a contribution. The method might not be generalizable to other complicated tasks. I would suggest that authors should prove that the method can be generalized to more complicated tasks.

---

> ### Author Response · Authors · 2021-08-31
> **Review of Paper85 by Reviewer teuP Rebuttal**
>
> We want to thank the reviewer for their comments and suggestions and have adapted the paper to address their concerns. We answer the main comments below:
>
> 1. There is no comparison with other methods.
>
> R: We clarify that there are no other methods approaching this problem, other than MIQP (which is explored in the paper). Since there is no specific work dealing with this problem, we compare our approach with variations on the network structure, under different levels of complexity. In Section 5, under Architectures we reference works that are similar and fall in the same category of structure at the various levels we explore.
>
> 2. The method also requires a differentiable simulator because the cost function is based on the simulator. It might work for these 2D examples. But it might not be easy to have a differentiable simulation for more complicated tasks.
>
> R: We added text to the paper describing how there exist approaches for 3D differentiable simulation which work well in large-scale problems.
>
> 3. Is the model (two parts) trained end-to-end or separately?
>
> R: We explain in section 5 (architectures paragraph) how all the training is performed for each different architecture.
>
> 4. In the first row of Fig.7, two fingers are on one side of the object. The question is why the object can be raised.
>
> R: The reviewer is correct at pointing out this behavior. However, it is physically consistent with the presence of friction on each side of the object. Note how the object has a top concavity where force is also applied, hence lifting the object.
>
> 5. According to the authors themselves, the method cannot be easily generalized to 3D tasks
>
> R: We clarified our conclusion to better describe how this method can be generalized to 3D, since parameters can be obtained in 3D as well and segmented point clouds can serve as input to the model. Moreover, we added text to the paper describing how there exist approaches for 3D differentiable simulation which work well for large-scale problems.

---

> > ### Comment · Reviewer_teuP · 2021-08-31
> > **I will keep my old score.**
> >
> > Thank you for the authors' answer.
> >
> > The key point of the method is to solve an inverse problem that produces the parameters set that is used either directly for supervised learning (L_p) or for QP (L_QP and L_sim). The model-based part provides different costs for learning the first part. The experiments are to evaluate which cost (or the combination) is the best.
> >
> > My main concern is whether it can be extended to the 3D or the real task. As I mentioned in my review, this parameter set is too simple in the examples mentioned in the paper (considering only gravity and 2D frictions). If it increases rapidly, the problem becomes more complicated. It can also be ill-posed like other inverse problems and can have multiple solutions, hence, leads to the average problem especially with a simple forward structure. The authors did not consider this problem explicitly and tried to use a simple forward NN architecture to solve this inverse problem.
> >
> > The contribution is different cost functions in the paper. I am not sure how these cost functions can help in this case. Experimentally, the authors proved that their method works well. However, it might be the reason that the target task is simple. So, I suggest that the authors can try the real task.

---

> > > ### Author Response · Authors · 2021-08-31
> > > **Review of Paper85 by Reviewer teuP Rebuttal 2**
> > >
> > > We thank the reviewer for their comments and advice on how to improve the paper. We specially thank that they find “The paper is well written” and that “The model seems to be promising”. We want to add a few clarifications:
> > >
> > > 1. To the statement: “My main concern is whether it can be extended to the 3D or the real task. As I mentioned in my review, this parameter set is too simple in the examples mentioned in the paper (considering only gravity and 2D frictions)
> > >
> > >     a. The updated section 6 describes how this method can be extended to 3D. We want to clarify that the parameter set used in this paper would remain the same if we were to pose a 3D manipulation problem, since spatial mechanics are also described by friction cones, facets and external contacts (see [11,22,23].
> > >
> > >     b. In fact, there exist many tools that can be applied to generate data and implement our pipeline in a 3D setting, see [6, 8, 31].
> > >
> > >     c. We recognize the simplicity of our approach, focused on a toy-version of the problem, and are very upfront about this in the paper. However, we stress how a 2D formulation is more appropriate towards understanding the impact of such a novel framework. Testing this approach in a real-world setting or with 3D data is out of the scope of our contribution and likely would obfuscate an appropriate analysis of this pipeline.
> > >
> > > 2. To the statement: “can also be ill-posed like other inverse problems and can have multiple solutions, hence, leads to the average problem especially with a simple forward structure. The authors did not consider this problem explicitly"
> > >
> > >     a. We thank the reviewer for bringing this point forward. However, this statement is factually incorrect since the paper devotes section 4.3 describing how we deal with multiple solutions when training and evaluating this method by applying simulation.
> > >
> > >     b. We explicitly introduce the loss function L_{Sim} as a solution to solve this problem end-to-end without being ill-posed.
> > >
> > > 3. To the statement “The contribution is different cost functions in the paper“
> > >
> > >     a. Our updated introduction better explains how the contribution of this paper lies in the application of differentiable optimization and differentiable simulation (via finite differences) as elements that can aid the inference of non-prehensile manipulation skills. This is enabled by our proposed formulation of inverse dynamics and smoothed contact mechanics for simulation.

---

### Official Review · Reviewer_XGL2 · 2021-07-26

**Originality:** Good
**Technical Quality:** Very Good
**Clarity Of Presentation:** Good
**Impact:** 4

**Recommendation:**

Weak Accept: I recommend accepting the paper, but will not argue for my recommendation if the majority of other reviewers have a different opinion.

**Summary:**

The paper presents a pipeline to go from 2D video input of physical interactions to a set of mechanical parameters that explains the sequence (via a learned LSTM model) and subsequently uses the parameters to optimize for finger manipulation actions and physically simulate resulting object trajectories. A number of ablations and comparisons are conducted with losses imposed on object trajectories, finger trajectories, and/or the mechanical parameters. Compared to purely learned architectures the approach shows favorable performance.


**Issues:**

- How exactly is the dataset generated? Can this be described in more detail in the appendix?
- Multiple times the simulator is called “differentiable” when in fact finite differences are used. I’m sure this might technically still be the correct terminology but I think nowadays “differentiable simulation” is most often understood as automatic differentiation. Especially, since the “finite difference” part is only revealed in the limitation section at the end of the paper, this feels slightly misleading.


**Reviewer Expertise:**

Good: General knowledge of the area

**Strengths And Weaknesses:**

Strengths:
- The presented problem is general and highly relevant for robotics.
- The paper evaluates a lot of variations, ranging from purely learning based pipelines to those that interweave model-based components.
Limitations section

Weaknesses:
- The “visual” data part of the method seems a little bit exaggerated. The input data doesn’t seem to have a lot of the characteristics of real data coming from a camera sensor. The input layer could equally be interpreted as an imposed representation rather than a raw measurement.  Is there any reason the approach could not be evaluated e.g. on the OmniPush dataset?
- In its current form the paper is a little bit demanding when it comes to trying to extract insights/lessons learned. There is a lot to unpack, ranging from loss functions to architecture structures, run-time etc. Maybe re-structuring the Experiment section into a number of tested hypotheses could clarify this.


**Summary Of Recommendation:**

My main criticism (making the acceptance “weak”) is due to the nature of the data: It is not only limited to the plane but more importantly not based on real images.

---

> ### Author Response · Authors · 2021-08-31
> **Review of Paper85 by Reviewer XGL2 Rebuttal**
>
> We want to thank the reviewer for their comments and suggestions and have adapted the paper to address their concerns. We answer the main comments below:
>
> 1. The “visual” data part of the method seems a little bit exaggerated. The input data doesn’t seem to have a lot of the characteristics of real data coming from a camera sensor.
>
> R: We recognize how the title could be interpreted differently and clarified very upfront the nature of our data.
>
> 2. The input layer could equally be interpreted as an imposed representation rather than a raw measurement. Is there any reason the approach could not be evaluated e.g. on the OmniPush dataset?
>
> R: We agree with this comment and have added an Appendix B that shows how we can infer manipulation plans from tasks using object shapes from the OmniPush dataset (without any extra training). We segment the data manually and not from real-world video; however, this is a demonstration of how this framework performs when presented with data akin to the real world. We also describe how our goal is to have the input layer to be a segmented video coming from a real world setup.
>
> 3. In its current form the paper is a little bit demanding when it comes to trying to extract insights/lessons learned.
>
> R: We improved the introduction and conclusion to better extract our main insights.
>
> 4. How exactly is the dataset generated? Can this be described in more detail in the appendix?
>
> R: We added Appendix A that describes the data generation process in more detail.
>
> 5. Multiple times the simulator is called “differentiable” when in fact finite differences are used
>
> R: We recognize this can be misleading and we make sure to clarify the backward pass is done with finite-differences across the paper. We want to remark that the simulator is made differentiable by our smoothing of the contact-complementarity function \delta() with a sigmoid (see section 4.4). Without this approximation, the finite-differences approach would return no data.

---

### Official Review · Reviewer_5sxA · 2021-07-27

**Originality:** Good
**Technical Quality:** Very Good
**Clarity Of Presentation:** Good
**Impact:** 3

**Recommendation:**

Weak Accept: I recommend accepting the paper, but will not argue for my recommendation if the majority of other reviewers have a different opinion.

**Summary:**

This submission aims to identify robot actions that involve contacts to manipulate an object given a video of an object moving along a planar motion. To achieve this objective, it proposes an ML pipeline that decodes the video informed by contact mechanics and by using differentiable optimization and simulation. The pipeline has 3 differentiable steps: i) a neural model estimates mechanical parameters such as object shape, trajectory and allowable forces, ii) a convex optimization solver to find contact and force trajectories for the robot fingers, iii) a simulation that executes the robot actions. The first two steps are trained using supervision from a contact trajectory optimizer, while the last step is trained in an end-to-end fashion. The evaluation aims to show that the proposed pipeline outperforms ML approaches without knowledge of mechanics in terms of generalization to new objects and motions. It also argues computational efficiency against non-learning approaches, such as the contact trajectory optimizer.

**Issues:**

Highlighting the challenge of the simplified setup, better supporting some of its choices (e.g., free flying object in input) and where the state-of-the-art is in this domain relative to the contribution will help.

Is there a way to evaluate the impact of the approximations for the QP so that it is differentiable?

**Reviewer Expertise:**

Good: General knowledge of the area

**Strengths And Weaknesses:**

The combination of machine learning primitives and reasoning about the underlying mechanics, which this work pursues, is appreciated and is reasonable for the target domain. The self-supervision step by using the differentiable simulator and avoid the need for ground truth labels for parameters or actions is also a strength of the proposed framework.

At the same time, the paper limits the scope to: i) animated planar motions, i.e., the object’s poses exist in SE(2) poses and the visual data are very clear, there are no occlusions due to a manipulator (or human demonstration), and ii) simulated, freely-moving point-fingers. These choices - a toy setup as it is called by the authors - remove sensing or actuation noise and obviously significantly simplify the challenge. The paper mentions that “The main challenges ... come from interpreting video data.. to extract an implicit representation of the object shape and the motion of the task” but one is left wondering how hard of a challenge this is given the animated, clear data available.  For instance, the objects considered also feel quite artificial and disconnected from applications.

The work is emphasizing at multiple points that the input data do not show the actions performed to achieve the object motion as one of the challenges, since it can result in an ambiguity regarding the set of robot actions that result in the same object motion. But what is the real-world setup that motivates this type of input? One can consider human demonstrations but then the contact points are part of the visual input and the key challenge arises due to occlusions.

The computational benefits of the proposed approach were not clear as the backwards pass seems to result in significant computational overhead.

The qualitative results show that there is still a significant gap between the output of the proposed approach and the desired object motion.

The authors should consider better highlighting in Figure 5 (right) and in the “Architectures” section the incremental relationship of MDR, CVX and DDM (which ends up being the final proposed approach) as well as NN and NNM.

Some minor issues:
Line 120: “Hence, The first”
Line 141: “QP it can be”
Line  143: “QP must be have”
Line 194: “objects objects”
Line 224: “with with”
Line 245: “as the our model”

**Summary Of Recommendation:**

It is understandable that certain compromises in the realism of the challenge are considered in order to explore this interesting space. The reviewer was ambivalent on whether these compromises in the setup still allow us to be informed about the type of solutions that will address realistic problems or whether the gap is significant enough that this is not possible. Overall, however, it will be beneficial for those working in dexterous manipulation to learn about this differentiable pipeline and what it can achieve even in this simple setup.

---

> ### Author Response · Authors · 2021-08-31
> **Review of Paper85 by Reviewer 5sxA Rebuttal**
>
> We want to thank the reviewer for their comments and suggestions and have adapted the paper to address their concerns. We answer the main comments below:
>
> 1. The computational benefits of the proposed approach were not clear as the backward pass seems to result in significant computational overhead.
>
> R: We clarify that the backward pass is only performed while training the network, not during inference. Hence, one should look solely at the forward pass when comparing the planning speed compared to MIQP.
>
> 2. The authors should consider better highlighting in Figure 5 (right) and in the “Architectures” section the incremental relationship of MDR, CVX, and DDM (which ends up being the final proposed approach) as well as NN and NNM.
>
> R: We modified Fig. 5 to better highlight the incremental relationship between these networks.
>
> 3. Highlighting the challenge of the simplified setup, better supporting some of its choices (e.g., a free-flying object in input) and where the state-of-the-art is in this domain relative to the contribution will help.
>
> R: We added text describing how a free-flying object represents a task specification that could be obtained through video prediction or non-occlusive demonstrations. We also added text describing how the state of the art in model-based manipulation is actively focused on setups similar to ours, due to the challenges associated with hybrid mechanics and friction.
>
> 4. Is there a way to evaluate the impact of the approximations for the QP so that it is differentiable?
>
> R: We added text to the paper that describes how the slack variable \epsilon value is minimized at each training iteration and set to zero after training, leading to an exact QP.

---

### Meta-Review · Area_Chair_k7r3 · 2021-08-11

**Recommendation:** Accept (Poster)
**Confidence:** 4

**Metareview:**

Strengths:

- The proposed method combines learning and planning aspects in an interesting way. The combination of neural network methods with convex optimization methods was appreciated.

- The problem of learning contact rich manipulation from video is important.

- Comprehensive ablations.

Weaknesses:

- All the reviewers identified the simplistic evaluation as a major weakness, specifically the visually simplistic images and the planar manipulation setting. A successful evaluation of the method using more realistic video would make the work much more impactful (one reviewer suggests the OmniPush dataset).

- It is not clear that the qualitative performance demonstrated in fig 7 warrants the (apparently) significant computational overhead involved here and the overall complexity of the method.

Post-rebuttal:

Significant concerns remain about whether this approach will scale up to moderately realistic manipulation problems. However, the approach is very novel and executed well and is probably of interest to a significant body of researchers in the field.

---

> ### Author Response · Authors · 2021-08-31
> **Meta-review rebuttal**
>
> We thank all reviewers and the area chair for their thoughtful feedback and comments. We appreciate that they found our approach interesting/general/relevant/promising (k7r3, 5sxA, XGL2, teuP), paper well written (teuP) and experiments/ablations comprehensive (k7r3, XGL2), and view self-supervision via simulation as a strength (5sxA). We address the various points raised by the reviewers in the individual responses. Here we address the two main weaknesses summarized in the meta-review.
>
> 1. We would like to stress and reiterate that our primary focus in this work is to evaluate hybrid architectures that combine the strengths of both model-based and deep learning techniques for manipulation tasks. There is limited existing work in this domain and our specific instantiation of a fully differentiable and structured pipeline is a novel direction. Therefore, as we state in the paper, we limited the problem setup to planar manipulation tasks that are fairly common in robot manipulation literature even today, particularly in the context of non-prehensile manipulation. Additionally, we limit the input to animated videos since this allows us to focus on the manipulation part of the problem and not confound the problem complexity with vision-related challenges like 3D scene understanding, hand-object interaction, occlusion, etc that are also active areas of research (note that it is also fairly common to reduce complexity by using AR tags, etc). Our hope here is to simultaneously make advances on the manipulation side to be able to leverage such vision techniques which primarily tend to rely on deep learning, thus making our current work poignant. We do acknowledge the above two points as limitations in the paper and will add this discussion. We would encourage the reviewers to consider that this work is the first step in this direction and we think it will be valuable for the robotics community to learn about our hybrid and fully differentiable architecture.
>
> 2. The computational overhead currently comes primarily from differentiating through the simulator and is only incurred during training and *not* at inference time. There is a growing body of active work in differentiable simulation and we believe as this area matures the capabilities and the computation speeds will improve. The main comparison of our method is w.r.t. model-free approaches (NN / NNM) where we show our approach does outperform them both quantitatively and qualitatively. In comparison to model-based approaches (we improve on the inference time), it is important to note that these assume having access to privileged information like object shape and trajectory.
>
> We have updated the paper in order to address each these and other reviewer points, added Appendix A describing dataset generation and Appendix B that includes new qualitative results that demonstrate how our approach can be applied to more realistic shapes from the OmniPush dataset.

---

### Decision · Program_Chairs · 2021-09-13

**Decision:**

Accept (Poster)

**Comment:**

Strengths:

- The proposed method combines learning and planning aspects in an interesting way. The combination of neural network methods with convex optimization methods was appreciated.

- The problem of learning contact rich manipulation from video is important.

- Comprehensive ablations.

Weaknesses:

- All the reviewers identified the simplistic evaluation as a major weakness, specifically the visually simplistic images and the planar manipulation setting. A successful evaluation of the method using more realistic video would make the work much more impactful (one reviewer suggests the OmniPush dataset).

- It is not clear that the qualitative performance demonstrated in fig 7 warrants the (apparently) significant computational overhead involved here and the overall complexity of the method.

Post-rebuttal:

Significant concerns remain about whether this approach will scale up to moderately realistic manipulation problems. However, the approach is very novel and executed well and is probably of interest to a significant body of researchers in the field.